# Clinical Characteristics and Multimodal Imaging Findings of Central Serous Chorioretinopathy in Women versus Men

**DOI:** 10.3390/jcm11061706

**Published:** 2022-03-19

**Authors:** Elodie Bousquet, Héloïse Torres-Villaros, Julien Provost, Martine Elalouf, Anthony Gigon, Irmela Mantel, Aurélie Timsit, Francine Behar-Cohen

**Affiliations:** 1Department of Ophthalmology, Ophtalmopôle, Hôpital Cochin, Assistance Publique-Hôpitaux de Paris, AP-HP, Université de Paris, 75014 Paris, France; heloise.torres.villaros@gmail.com (H.T.-V.); julien0provost@gmail.com (J.P.); timsit.aurelie@gmail.com (A.T.); francine.behar@gmail.com (F.B.-C.); 2Centre de Recherche des Cordeliers, Université de Paris, 75006 Paris, France; 3Department of Ophthalmology, Jules Gonin Eye Hospital, Fondation Asile des Aveugles, University of Lausanne, 1015 Lausanne, Switzerland; martine.elalouf@hispeed.ch (M.E.); anthony.gigon@fa2.ch (A.G.); irmela.mantel@fa2.ch (I.M.)

**Keywords:** central serous chorioretinopathy, epitheliopathy, gravitational tracks, macular neovascularization, women, pachychoroid neovasculopathy

## Abstract

(1) The aim of this study was to compare the clinical characteristics and multimodal imaging findings of central serous chorioretinopathy (CSCR) between women and men. (2) Women and men with CSCR were compared in terms of their age and risk factors, the clinical form of their disease, multimodal imaging findings and the presence of macular neovascularization (MNV) on optical coherence tomography (OCT)-angiography. (3) Results: The data of 75 women and 75 men were compared. The women were significantly older than the men (52.2 years versus 45.7 years; *p* < 0.001). Corticosteroid intake was more frequent in the women (56% versus 40%; *p* = 0.05). The women had a single foveal subretinal detachment more often than the men (73.3% versus 46.9%; *p* < 0.001) and they often had fewer gravitational tracks (16.3% versus 29.6%; *p* = 0.03). On mid-phase indocyanine green angiography, hyperfluorescent plaques were detected less often in the women than in the men (48% versus 72.2%, *p* = 0.001). MNV was detected on OCT-angiography in 35.9% of the women and in 13.3% of the men (*p* = 0.004). (4) In the women, CSCR occurs at an older age, is more often unifocal foveolar, and is associated with a higher rate of MNV. The reasons for these gender-related differences remain to be determined.

## 1. Introduction

Central serous chorioretinopathy (CSCR) is a chorioretinal disease characterized by the presence of serous retinal detachment (SRD) associated with retinal pigment epithelium (RPE) detachment (PED) and a thick choroid [1]. CSCR usually occurs in middle-aged men [1]. Indeed, the annual incidence of CSCR in an Olmstead County, Minnesota population study was 9.9 and 1.7 per 100,000 individuals in men and women, respectively [2]. In addition, at the onset of the disease, women are older than men [3]. This finding was confirmed in a Japanese cohort of 147 CSCR patients, in which the phenotype was stratified by age, and the proportion of women was higher in the age group > 50 years [4].

To date, several risk factors for CSCR have been identified. The most commonly reported risk factor is exposure to systemic corticosteroids [5]. However, in corticosteroid-induced CSCR, male predominance is less obvious [6]. Several other predisposing or contributing factors have been identified [1,7], including the presence of psychiatric disorders, particularly depression [8] and stress, hypertension, cardiovascular disease, sleep apnea [9], shift work [10], allergic disorders, helicobacter pylori infection [11], and genetic risk factors [1,12]. 

The pathogenic mechanisms of CSCR and its risk factors are still debated. Pachychoroid has been widely described as a predisposing phenotype [13,14] and, more recently, choroidal venous overload has emerged as a new hypothesis [15]. However, the causes of these choroidal vascular deregulations remain unknown and their relationship with the male predisposition is unclear. The hypothesis that the overactivity of the mineralocorticoid receptor (MR) pathway in the retina and/or choroid could contribute to CSCR has emerged from animal models [16,17]. It could also help to explain the male predisposition, since progesterone is a known antagonist of the MR pathway [18], which could limit its inappropriate activation in premenopausal women.

In addition to gender differences in terms of prevalence, the disease phenotype might be different. However, because of the scarcity of the disease in women, few studies have characterized the specificities of their clinical forms [3,19,20]. It was shown that women were older at the onset of the disease [3,19] and had less diffuse RPE damage than men [20].

The aim of this study was to compare the clinical characteristics and multimodal imaging findings between women and men with CSCR.

## 2. Materials and Methods

### 2.1. Ethics Statement

The study was approved by the Ethics Committee of the French Society of Ophthalmology (IRB 00008855 Société Française d’Ophtalmologie IRB#1) and of the Swiss Federal Department of Health (CER-VD 2017-00493). The study adhered to the tenets of the Declaration of Helsinki (1964).

### 2.2. Study Design

This was a retrospective study conducted in the departments of Ophthalmology of Cochin Hospital, Paris, France and Jules Gonin Eye Hospital in Lausanne, Switzerland between 2012 and 2020.

### 2.3. Study Patients

The medical records and imaging findings of female patients with CSCR with a follow-up of at least 3 months were reviewed. A control group of consecutive men with CSCR was included. Exclusion criteria were: (1) Presence of any other retinal disease, especially age-related macular degeneration (AMD) or significant drusen in the posterior pole, dome-shaped macula in case of high myopia, diabetic retinopathy, and vitreomacular traction; (2) a follow-up <3 months; and (3) poor image quality. 

### 2.4. Study Protocol

The medical data collected included age, gender, history of corticosteroid intake, and the best-corrected visual acuity (BCVA) converted into logarithm of the minimum angle of resolution (LogMAR). For the female group, the menopausal and pregnancy statuses were also recorded. Bilateral CSCR was defined as the occurrence of a serous retinal detachment (SRD) in both eyes. Acute CSCR was defined as a first episode of SRD without widespread RPE dysfunction or gravitational tracks. In case of relapsing SRD, CSCR was classified as recurrent. Complex CSCRs (or persistent/chronic forms) are defined as all the other forms, which means CSCR with persistent SRD and/or widespread/multifocal RPE alterations. Multimodal imaging was performed and included spectral-domain optical coherence tomography (SD-OCT; Heidelberg Spectralis Engineering, Heidelberg, Germany), blue fundus autofluorescence (BAF), fluorescein angiography and indocyanine green angiography (FA and ICGA, Spectralis, Spectralis Engineering, Heidelberg, Germany) and OCT-angiography (Angiovue, Optovue, Fremont, CA, USA).

### 2.5. Image Analyzes

The routine clinical acquisition protocol for OCT included a volume scan centered on the fovea in enhanced-depth-imaging (EDI) mode. The OCT cubes were analyzed to determine the location of the SRDs. The SRDs were classified as “unifocal foveal SRD” in case of single macular SRDs or as “multifocal SRD” when two or more SRDs were detected (Figure 1). The presence of a PED was assessed on the entire OCT cube. The form of the PED was also analyzed and classified as a dome-shaped or flat irregular PED (FIPED), defined by an irregular elevation of the RPE [21]. The choroidal thickness was manually measured on the horizontal EDI-OCT B-scans passing through the fovea. The presence of a multifocal or unifocal hyper/hypoautofluorescent area and gravitational tracks was assessed on BAF. The presence of focal leakage with a smokestack or inkblot pattern typical of CSCR was assessed on FA. The presence of mid-phase hyperfluorescent plaques (MPHP) on ICGA was recorded. OCT-angiography was used to analyze the presence of macular neovascularization (MNV). Both 3 × 3 mm and 6 × 6 mm OCT-A volume scans were captured for each eye. For MNV detection, the outer retinal and the choriocapillaris segmentation slabs were analyzed with manual adjustment of the segmentation boundaries when necessary to detect decorrelation signals suggestive of neovascularization. All the imaging analyses were performed by two trained retina specialists (HV and EB). In case of disagreement, a third retina specialist (FBC) analyzed the images. 

### 2.6. Statistical Analysis

The descriptive data are presented as the mean ± standard deviation (SD) for quantitative variables and as counts and percentages for categorical variables. The comparisons between variables were performed using a Mann–Whitney test or a Chi-squared test with or without Yates continuity correction, as appropriate. The statistical analyses were performed using Xlstat software (version 2020; Addinsoft, Paris, France). All the *p* values were two-sided and *p* values ≤ 0.05 were considered statistically significant. 

## 3. Results

### 3.1. Patients’ Demographics and Clinical Form of CSCR

The study included 184 eyes (86 women’s eyes and 98 men’s eyes) from 150 patients (75 women and 75 men). The patients’ demographics and the clinical forms of their CSCRs are summarized in Table 1.

The mean follow-up duration was 33.2 ± 27.8 months, without any significant difference between the men and women.

The mean age at the time of presentation was 52.2 ± 11.6 years in the women and 45.7 ± 8.9 years in the men (*p* < 0.001). Four women (5.3%) were pregnant and 43 women (57.3%) were postmenopausal. Previous corticosteroid intake was more often reported in the women than in the men (54.7% versus 40%; *p* = 0.05).

CSCR was less frequently bilateral in the women than in the men (16% versus 34.7%; *p* = 0.009). An acute or recurrent form of CSCR was found in 27.9% of the women and in 26.5% of the men (*p* = 0.83). Thus, the frequency of persistent/chronic forms was similar between the women and men in our cohort.

### 3.2. Clinical and Imaging Findings 

The baseline BCVA was not different between the women and the men. The multimodal imaging findings are summarized in Table 2. The subfoveal choroidal thickness was significantly lower in the women than in the men (432.4 ± 104.2 µm versus 473.8 ± 83.7 µm; *p* = 0.008). The SRD and the RPE damage was less often multifocal in the women than in the men (*p* < 0.001 and *p* = 0.009, respectively). Regarding the location of the SRDs and epitheliopathy, the women had a single foveal subretinal detachment in 73.3% of cases and they had gravitational tracks less frequently compared to the men (16.3% versus 29.6%; *p* = 0.03). 

On FA, a similar rate of focal leakage was found between the women and the men. However, on mid-phase ICGA, hyperfluorescent plaques of choroidal hyperpermeability were less frequently detected in the women than in the men (48% versus 72.2%, *p* = 0.001). OCT-angiography was available for 124 eyes (64 women and 60 men). It allowed the identification of type-1 MNV in 35.9% of the women and in the 13.3% of men (*p* = 0.004). 

Overall, in our cohort, the most common phenotype of CSCR found in the women was a single unilateral circumscribed foveal subretinal detachment (Figure 2, which was frequently associated with type 1 MNV.

## 4. Discussion

It is unclear whether CSCR has the same clinical phenotype in men and women, although this question is important for diagnosis, prognosis and genetic studies. In our series of patients, we confirmed, as previously described [3,4], that the age of onset is higher in women than in men, although only 57% of the women in our study were postmenopausal. By comparison, Perkins et al. [19] reported a rate of 44% of post-menopausal women in a study of78 women with CSCR. Hormonal status therefore might not be the only factor influencing the age of CSCR onset in women. The exact role of sex hormones in CSCR is unclear. Whether male sex hormones promote the occurrence of CSCR, female sex hormones protect from CSCR or both are involved remains to be determined. Measurements of serum testosterone levels in CSCR patients have shown conflicting results. Two studies failed to find any difference in serum testosterone levels between patients with CSCR and controls [22,23], while one study showed an increased level in CSCR patients [24]. Progesterone antagonizes MR activation through aldosterone in humans [18] and could also antagonize the effect of the glucocorticoid receptors in specific cells [25], while the estrogen receptor inhibits the transcriptional regulatory function of the MR [26]. Sex hormones could therefore significantly influence the balance of glucocorticoid receptor/MR activation during both aging and pregnancy. However, the exact effect of sex hormones on the RPE/choroid remains to be investigated.

Another statistically significant result of our study was that CSCR was more often unilateral in the women than in the men. We found bilateral CSCR in 16% of the women in our cohort. This rate is consistent with a previous report by Quillen et al. [3]. It could be assumed that bilateralization occurs over time and that it may be observed after a longer disease duration and with aging. Since multifocal subretinal detachment and extended epitheliopathy were more frequently observed in the men, asymptomatic episodes could have occurred more frequently at a younger age in the men, explaining why bilateral disease was more frequently observed at the time of diagnosis in the men.

In our study, the typical form observed in our female population was a single foveal subretinal detachment with no other location or signs of a previous episode outside the fovea. These findings are in agreement with the study by Hanumunthadu et al. [20], who recently described higher rates of diffuse RPE alterations, diffuse leakage and RPE tracts in men. In both populations, women seem to have less diffuse RPE alterations and a unifocal form of the disease involving predominantly the foveolar area. Interestingly, the rate of hyperfluorescent plaques corresponding to choroidal hyperpermeability on mid-phase angiography was lower in women than in men. We have recently shown that these plaques could correspond to an early sign of epitheliopathy secondary to choroidal hyperpermeability [27]. Altogether, these findings demonstrated that RPE damage is less extensive in women than in men.

However, despite the fact that the disease was unifocal, with minimal diffuse epitheliopathy, the rate of choroidal neovascularization was higher in the women (36% versus 13% in the men). This result is consistent with the study by Shiragami C et al. [28], who assessed 363 eyes with CSCR. The female gender, a chronic form of the disease and poor visual acuity at baseline were identified as risk factors for MNV [28]. Siedlecki et al. proposed a new classification of pachychoroid spectrum diseases, considering that type-1 MNV associated with CSCR should be referred to as “pachychoroid neovasculopathy” [29]. Thus, using this definition, we could infer that 36% of the women and 13% of the men in our cohort had pachychoroid neovasculopathy. More importantly, the visual acuity in the women was not worse than in the men despite the high prevalence of MNV among the women and their older age.

This study has some limitations, including its retrospective design, the variable duration of the follow-up and some missing data, mainly concerning the OCT-angiography findings available in 2016 in our centers. We did not evaluate the systemic circulatory dynamics, nor the scleral thickness, that could have explained the differential CSCR phenotypes observed between the men and women. This should be explored in future studies. Furthermore, the low rate of acute/simple CSCR forms found in the women and men in our cohort could be related to the fact that most of the cases in our tertiary referral center are more severe forms, limiting the inclusion of the simple and spontaneously resolved cases. Regarding the fact that the control group consisted of men, we only included consecutive patients to avoid any selection bias.

## 5. Conclusions

In summary, in women, CSCR occurs at an older age; it is more often unilateral, unifocal foveolar and associated with a higher rate of type-1 MNV and a lower rate of diffuse RPE damage. Further studies are needed to evaluate the long-term prognosis for affected eyes and to determine whether visual acuity remains stable over time. It remains to be determined whether the genetic risk factors are identical in women and men. The differential diagnosis between CSCR and AMD may be challenging in these patients and multimodal imaging is needed to determine the optimal treatment option.

## Figures and Tables

**Figure 1 jcm-11-01706-f001:**
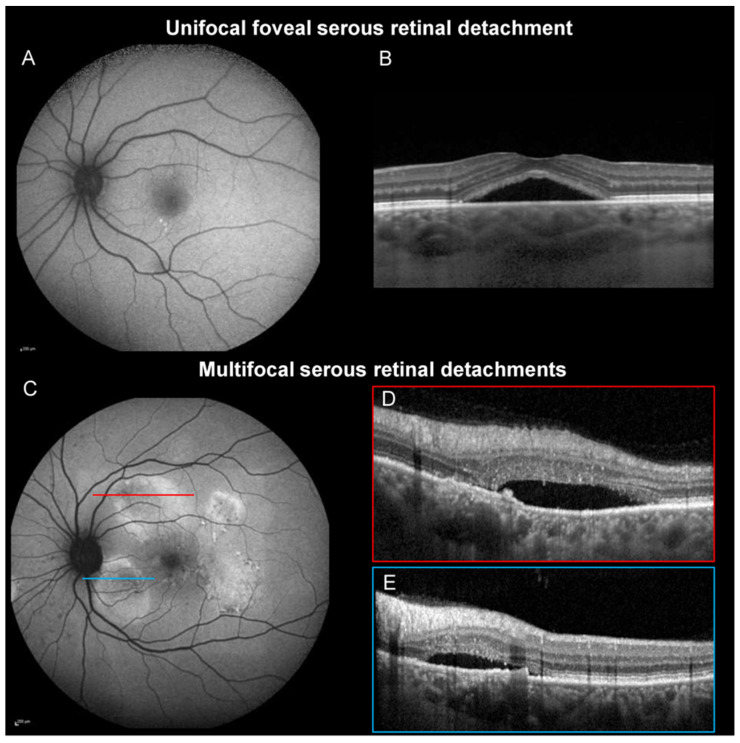
(**A**,**B**) Central serous chorioretinopathy (CSCR) with a unifocal macular serous retinal detachment (SRD). (**A**) Blue-light fundus autofluorescence (BAF) shows no signs of previous extra-macular SRD. (**B**) The OCT B-scan shows a macular SRD. (**C**,**E**) CSCR with multifocal SRD.(**C**) BAF shows a mixed multifocal round area of hyper/hypo-autofluorescence consistent with an active or resolved SRD. (**D**,**E**) The OCT B-scan passing through the round hyper-autofluorescent area shows subretinal detachments located superior to the fovea (**D**) and inferior to the optic disc (**E**).

**Figure 2 jcm-11-01706-f002:**
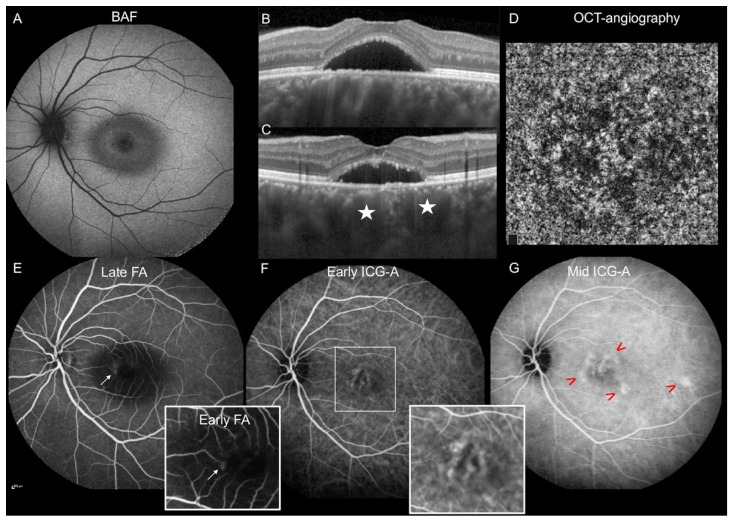
Multimodal imaging of a 34-year-old woman with central serous chorioretinopathy in the left eye. (**A**) Blue-light fundus autofluorescence shows a mixed hyper- and hypo-autofluorescent area at the macular serous retinal detachment (SRD). (**B**,**C**) The horizontal (**B**) and vertical (**C**) enhanced-depth-imaging (EDI)-OCT scans centered on the fovea show a macular SRD associated with dilated choroidal vessels (stars). (**D**) OCT-angiography at the level of the choriocapillaris shows no macular neovascularization. (**E**) Late-phase fluorescein angiography (FA) shows one focal leak (arrow). Insert: early-phase FA showing the focal leak (arrow). (**F**) Early-phase indocyanine green angiography (ICGA) shows dilated macular choroidal veins magnified in the insert. (**G**) Mid-phase ICGA shows multifocal hyperfluorescent plaques (arrrowheads).

**Table 1 jcm-11-01706-t001:** Comparison of patients’ demographics and the clinical forms of central serous chorioretinopathy, according to gender.

	Women (*n* = 75 Patients, 86 Eyes)	Men (*n* = 75 Patients, 98 Eyes)	*p* Value
Follow-up, mean ± SD, months	33.7 ± 25.4	32.7 ± 30.2	0.48 *
Age, mean ± SD, years	52.2 ± 11.6	45.7 ± 8.9	**<0.001 ***
Pregnancy, *n* (%)	4 (5.3%)		
Menopause, *n* (%)	43 (57.3%)		
Corticosteroid intake, *n* (%)	42 (56%)	30 (40%)	0.05 ^†^
Bilateral CSCR, *n* (%)	12 (16%)	26 (34.7%)	**0.009** ^†^
Acute/recurrent CSCR, *n* (%)	24 (27.9%)	26 (26.5%)	0.83 ^†^
Complex (persistent/chronic) CSCR, *n* (%)	62 (72.1%)	72 (73.5%)	

SD: standard deviation; CSCR: central serous chorioretinopathy.* Mann–Whitney test; ^†^ Chi-squared test.

**Table 2 jcm-11-01706-t002:** Comparison of the clinical characteristics and multimodal imaging findings of patients with central serous chorioretinopathy, according to gender.

	Women (*n* = 75 Patients, 86 Eyes)	Men (*n* = 75 Patients, 98 Eyes)	*p* Value
**Best-Corrected Visual Acuity at baseline,** logMAR, mean ± SD (Snellen)	0.21 ± 0.24 (20/32)	0.22 ± 0.3 (20/32)	0.42 *
**Spherical equivalent **** mean ± SD	0.6 ± 1.8	0.3 ± 1.2	**0.04** *
**OCT findings**			
Choroidal thickness (µm), mean ± SD	432.4 ± 104.2	473.8 ± 83.7	**0.008** *
Pigment epithelium detachment, *n* (%)			
At least one dome-shaped PED	19 (22.1%)	25 (25.5%)	0.59 ^†^
At least one flat irregular PED	59 (68.6%)	55 (56.1%)	0.08 ^†^
Unifocal foveal SRD, *n* (%)	63 (73.3%)	46 (46.9%)	**<0.001** ^†^
Multifocal SRDs, *n* (%)	16 (18.6%)	43 (43.9%)	**<0.001** ^†^
**Autofluorescence findings**			
Multifocal hyper-/hypo-autofluorescent area showing RPE damage, *n* (%)	36 (41.9%)	60 (61.2%)	**0.009** ^†^
Gravitational tracks, *n* (%)	14 (16.3%)	29 (29.6%)	**0.03** ^†^
**Fluorescein angiography**			
Focal leakage, *n* (%)	42 (56.8%)	54 (57.5%)	0.9 ^†^
**Indocyanine green angiography ^‡^**			
Hyperfluorescent plaques during the mid-phase, *n* (%)	36 (48%)	65 (72.2%)	**0.001** ^†^
**OCT-angiography *****			
Type 1 macular neovascularization, *n* (%)	23/64 (35.9%)	8/60 (13.3%)	**0.004** ^†^

SD: standard deviation; PED: pigment epithelium detachment; SRD: serous retinal detachment.* Mann–Whitney test; ^†^ Chi-squared test; ^‡^ ICG angiography available for 165 eyes; ** in phakic patients before development of cataract; *** OCT-angiography available for 124 eyes.

## Data Availability

Not applicable.

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
