# Peer review of "Clinical Characteristics and Multimodal Imaging Findings of Central Serous Chorioretinopathy in Women versus Men"

_jcm, 2022, doi:10.3390/jcm11061706_

Round 1
Reviewer 1 Report
â‘ In this study, optical coherence tomography angiography was performed on patients with central serous chorioretinopahty (CSC) to evaluate the presence of macular neovascularization (MNV) as pachychoroid neovasculovaphy (PNV). The results showed that MNV was detected in 35.9% of females and 13.3% of males.
Although CSC and PNV are the same disease spectrum, they should be differentiated from each other, as some studies have used PNV eligibility criteria.
However, in this study, authors considered cases with MNV detected as CSC associated with PNV. Therefore, if the clinical characteristics of CSC are to be the subject of investigation, it is necessary to exclude cases with PNV.
<References>
1) Miyake M, Ooto S, Yamashiro K et al:Pachycho¬roid neovasculopathy and age-related macular de¬generation. Sci Rep 5:16204, 2015
2) Hata M, Yamashiro K, Ooto S et al:Intraocular Vascular Endothelial Growth Factor Levels in Pachychoroid Neovasculopathy and Neovascular Age-Related Macular Degeneration. Invest Oph¬thalmol Vis Sci 58:292-8, 2017
â‘¡The examination findings and results of acute CSC and chronic CSC are different. For example, atrophic lesions and decreased visual acuity are indicated by chronic CSC. Therefore, it is necessary to consider the clinical characteristics of acute CSC and chronic CSC separately. Also, how did authors define chronic CSC in this study?
â‘¢It has been reported that there is a relationship between CSC and systemic circulatory dynamics and/or ocular perfusion pressure. These data are also related to gender, age, and other factors. Do the authors have any data on the systemic and ocular circulation dynamics of CSC patients in this study?
â‘£The choroidal thickness is related to the ocular axial length and refractive power. It is also known that hyperopia (short axial length) is common in CSC. Therefore, if the clinical characteristics of CSC are to be shown, the authors need to present those data.
Author Response
Reviewer 1
In this study, optical coherence tomography angiography was performed on patients with central serous chorioretinopahty (CSC) to evaluate the presence of macular neovascularization (MNV) as pachychoroid neovasculovaphy (PNV). The results showed that MNV was detected in 35.9% of females and 13.3% of males.
Although CSC and PNV are the same disease spectrum, they should be differentiated from each other, as some studies have used PNV eligibility criteria.
However, in this study, authors considered cases with MNV detected as CSC associated with PNV. Therefore, if the clinical characteristics of CSC are to be the subject of investigation, it is necessary to exclude cases with PNV.
<References>
1) Miyake M, Ooto S, Yamashiro K et al:Pachycho¬roid neovasculopathy and age-related macular de¬generation. Sci Rep 5:16204, 2015
2) Hata M, Yamashiro K, Ooto S et al:Intraocular Vascular Endothelial Growth Factor Levels in Pachychoroid Neovasculopathy and Neovascular Age-Related Macular Degeneration. Invest Oph¬thalmol Vis Sci 58:292-8, 2017
We thank the reviewer for raising this important point. Although pachychoroid neovasculopathy has been described in 20151, there is currently no consensus about its exact definition and some authors considered that macular neovascularization secondary to CSCR should be renamed as pachychoroid neovasculopathy. However, the aim of our study was to compare the different phenotype of CSCR between men and women. We included in this study cases with a history of CSCR and detection of MNV during the follow-up. That is why the term MNV secondary to CSCR seemed more accurate.
â‘¡The examination findings and results of acute CSC and chronic CSC are different. For example, atrophic lesions and decreased visual acuity are indicated by chronic CSC. Therefore, it is necessary to consider the clinical characteristics of acute CSC and chronic CSC separately. Also, how did authors define chronic CSC in this study?
There is currently no consensus on the classification of CSCR.2 The term chronic CSC can refer to the duration of the serous retinal detachment or to the area of RPE damage. To avoid any confusion, we have avoided this term in our study. We found the same rate of acute/ recurrent CSC among men and women. There is probably a selection biais given that patients were included in a tertiary center, that’s why the rate of acute/recurrent CSC is low in our study.
To answer the question of the reviewer, we have now added in the table 1 the number of persistent/chronic/neovascularized CSCR
â‘¢It has been reported that there is a relationship between CSC and systemic circulatory dynamics and/or ocular perfusion pressure. These data are also related to gender, age, and other factors. Do the authors have any data on the systemic and ocular circulation dynamics of CSC patients in this study?
That is a very interesting question indeed. We did not evaluate the ocular pressure perfusion nor the intervortex choroidal vein anastomosis in this study but these parameters could explained the difference between men and women and could be the subject will be for future studies.
â‘£The choroidal thickness is related to the ocular axial length and refractive power. It is also known that hyperopia (short axial length) is common in CSC. Therefore, if the clinical characteristics of CSC are to be shown, the authors need to present those data.
To answer the reviewer comment, we have now added the spherical equivalent of phakic patients included in the study in the table 2 of the revised version of the manuscript. As previously described, CSC patients are slitghly hypermyopic. Spherical equivalent in women : 0.6 ± 1.8 and 0.3 ± 1.2 in men.

Reviewer 2 Report
The authors compared the clinical characteristics and multimodal imaging findings of CSCR between women and men. The etiology of CSCR is still elusive. This study might add some new insights to the difference of CSCR in women and men. The comments are as below:
- Line 105, Please add what scan pattern was used for OCT-angiography
- Line 125, among 184 eyes, how many eyes from men and women, respectively?
- Figure2D, please clarify which slab was used to detect MNV or no MNV.
- In discussion, please also try to explain the CSCR difference in women and men in terms of the sclera.
Author Response
Reviewer 2
The authors compared the clinical characteristics and multimodal imaging findings of CSCR between women and men. The etiology of CSCR is still elusive. This study might add some new insights to the difference of CSCR in women and men. The comments are as below:
- Line 105, Please add what scan pattern was used for OCT-angiography
We added this sentence in the revised version of the manuscript : “both 3X3mm and 6x6 mm OCT-A(Optovue RTVue XR Avanti) volume scans were captured for each eyes. For MNV detection, the outer retinal and the choriocapillaris layers were analyzed with manual adjustment of the segmentation boundaries when necessary to detect decorrelation signals suggestive of neovascularization.”
- Line 125, among 184 eyes, how many eyes from men and women, respectively?
We added in the revised version of the manuscript : 184 eyes (86 eyes from women and 98 eyes from men).
- Figure2D, please clarify which slab was used to detect MNV or no MNV.
We now added in Fig 2D legend : “OCT-angiography at the level of the choriocapillaris”.
- In discussion, please also try to explain the CSCR difference in women and men in terms of the sclera.
Thank you for this excellent comment. Indeed, differences in sclera thickness or composition could explain the difference of the CSCR phenotype between men and women. We did not assess the sclera thickness in this study but it is for sure a very interesting point to address in further investigations.
We added the absence of sclera assessment in the limitation part of the discussion as below :
“ the sclera thickness was not evaluated in this study but could be an interesting parameter to assess in further studies.”

Round 2
Reviewer 1 Report
Thank you for your thoughtful response to the peer review. I think it is a great study and very interesting.